# Intimate partner violence experience, support seeking and coping strategies among pregnant women in Southwestern Uganda

Eve Katushabe[1,2]*, John Baptist Asiimwe[3]*, John Bosco Ndinawe[1,4],
Editor Abeneitwe[2], Agnes Katusiime[2], Gladys Nakidde[2], Vincent Batwala[5]

1 Department of Community Health, Mbarara University of Science and Technology, Mbarara City, Uganda, 2 Department of Nursing, Bishop Stuart University, Mbarara City, Uganda, 3 School of Nursing and Midwifery, Aga Khan University, Uganda Campus, Kampala, Uganda, 4 Road to Care, Kampala, Uganda, 5 Directorate of Research and Graduate Training, Mbarara University of Science and Technology, Mbarara City, Uganda

* katushabeeve@yahoo.com, ekatushabe@hs.bsu.ac.ug (EK); john.asiimwe@aku.edu (JBA)

## Abstract

Intimate partner violence (IPV) during pregnancy remains a global health challenge. This study aimed to explore pregnant women's experiences of IPV, support seeking, and coping strategies in Southwestern Uganda. Pregnant women who had experienced IPV during pregnancy were purposively selected and completed in-depth interviews, with data saturation reached after 25 participants. Guided by the feminist theory, data were analyzed deductively and inductively using thematic analysis. Participating pregnant women were exposed to physical violence (e.g., the gravid abdomen being stepped on), psychological violence (e.g., verbal abuse), sexual violence (e.g., being forced into uncomfortable sexual positions), and financial control and manipulation (e.g., economic dependency used as a means of control). Pregnant women attributed IPV to intergenerational violence, unequal power dynamics in their households, and differences in traditional gender roles and social norms between men and women. Some pregnant women used active coping strategies to overcome effects of IPV, such as sharing their IPV experiences with a confidant. Other women accommodated abuse through passive coping strategies, such as keeping silent. Furthermore, identified barriers to seeking help after experiencing IPV included a lack of awareness, negative experiences with healthcare providers, partner dependence, and feelings of shame, guilt, and fear. This study's findings suggest stakeholders need to prioritize IPV screening, care, referral, and sensitization in healthcare facilities and communities. In addition, rules and regulations that protect the rights of IPV survivors should be strengthened, and perpetrators held accountable for their actions.

**Data availability statement:** All relevant data is within the paper.

**Funding:** The research work funding came from the Massachusetts General Hospital Center for Global Health's project entitled "First Mile: Powering the Academic Medical Center to Deliver Health care in the community in Uganda," supported through the Wyss Medical Foundation. The funders had no role in the study design, data collection, analysis, interpretation, or preparation of the manuscript. The recipient of the funding is Eve Katushabe, under check number Q0006883.

**Competing interests:** The authors have declared that no competing interests exist.

## Introduction

Intimate partner violence (IPV) is defined as actual or threatened physical, sexual, psychological, or emotional abuse by a current or former sexual partner [1]. The literature indicates that IPV can take various forms, including physical, sexual, psychological [2] and economic abuse [3–7]. Economic violence has gained increased attention alongside other forms of abuse. It involves controlling a victim's access to resources, thereby reducing their economic security and self-sufficiency [8]. Economic violence also impacts women and children by causing instability in the home, especially when the perpetrators neglect their responsibility as family providers [9,10].

IPV is a serious social and public health problem. A high proportion of women experience IPV irrespective of socioeconomic status, cultural or religious background, or level of education [11,12]. World Health Organization (WHO) global estimates indicate 27% of women aged 15–years and over have experienced IPV in their lifetime [13]. The prevalence of IPV in Sub-Saharan Africa ranges from 10.8% to 59.9% [14–16]. In Uganda, 40% of women in a relationship and 27.8% of pregnant women were reported to have experienced IPV [17,18]. Research suggests IPV has a recurring pattern in numerous communities in Uganda that persists across generations, making it difficult to break the IPV cycle [19]. Furthermore, some women are socialized to accept unequal rights and roles through being submissive to their husbands, not owning assets, and not engaging in productive employment [19]. Previous studies identified several risk factors for IPV during pregnancy, such as alcohol consumption [11,12,20], the husband's education level [12,20,21], women's age [22,23], women's decision making ability [24], prior history of IPV [23], residence [20], and household resources [25]. Cultural norms in patriarchal societies may influence these factors directly and indirectly.

[2,26,27],

Pregnant women are more likely to experience IPV, due to their increased vulnerability, dependency, and stressful life transition. This is because pregnancy can increase a woman's physical and emotional dependence on her partner due to financial demands, physical restrictions, and the need for support which may embolden abusive partners to assert control. Anxiety about money, raising a child, and shifting roles in relationships can also be triggered by pregnancy [28]

IPV during pregnancy has harmful effects on the health of the mother and the unborn child [7,29,30]. Previous studies showed that pregnant women who experienced IPV were at a high risk for suffering from chronic stress [7,31], spontaneous abortion, antepartum hemorrhage, trauma associated with IPV, experiencing feelings of helplessness, postponing their initial antenatal care contact, and contracting sexually transmitted infections [16,32,33]. These factors may lead to serious mental health problems such as substance abuse, depression, and suicidal thoughts. Potential effects of IPV on the baby include premature rupture of membranes, preterm birth, low birth weight, fetal injuries, and perinatal or neonatal mortality [7,34–37].

Pregnant women often have feelings of shame, fear, and stigma after experiencing IPV. Those emotions can lead to victims' isolation and concealment of their

experiences. However, involving confidants (e.g., friends) as a coping strategy may help women cope with IPV through the provision of emotional support [38–40]. The literature suggests that women who experience severe physical and emotional violence often seek support services to help them deal with life's traumatic events [40–42]. Previous studies reported that victims of IPV who adopted certain coping strategies had a reduced risk for experiencing depression [43,44]. However, different coping strategies may have both beneficial and detrimental effects on an individual's mental and physical well-being [45,46]. A range of strategies that women use to seek consolation, support, or safety following IPV has been identified. These strategies included engaging in distractions such as watching movies on television, singing, and chatting with friends, diverting their partner's focus, and resisting violence [42,45]. Other documented strategies comprised seeking social support, engaging in problem-solving, practicing self-reflection, committing to religious beliefs, reading Bible scripture, and consulting with pastors for guidance and prayer [46–51]. Other coping strategies have also been adopted by victims of IPV, such as fighting back, publicly disclosing the violent behavior to co-workers or friends, blaming themselves [48,49], acceptance, and using emotion-focused strategies (e.g., substance use) [48]. It is worth noting that coping strategies commonly practiced in African settings included self-reflection, religious devotion, and seeking social support. Unfortunately, these coping mechanisms do not address male IPV behaviors; rather, they encourage women to tolerate IPV. Most importantly, quantitative research methods have been used to identify the majority of these techniques among women in general.

A clear understanding of coping strategies used for IPV during pregnancy is necessary to develop focused IPV management interventions [52]. Pregnancy offers a distinct opportunity for healthcare providers to connect with pregnant women, screen for IPV, and intervene during prenatal care contacts [53]. Currently, the WHO encourages women to have eight contacts with care providers during pregnancy [54]. IPV screening is an integral part of antenatal care (ANC) for every woman. It improves pregnancy outcomes and allows maternal mental health conditions to be prevented and treated [54]. Early IPV screening means healthcare providers can intervene with counseling and other support services to prevent development of mental health conditions [55].

Although Uganda's Clinical Guidelines recommend IPV screening during prenatal care [56], there is insufficient documented information about pregnant women's qualitative accounts of IPV experiences, support-seeking behavior, and coping strategies. Most previous studies reported different forms of IPV experienced during pregnancy, their prevalence, and associated factors using quantitative methods [17,26]. Therefore, this qualitative study aimed to explore pregnant women's experiences of IPV, support seeking, and coping strategies in Southwestern Uganda to inform response plans for violence against women.

## Theoretical framework

Although many theoretical perspectives attempt to explain why women in general are victims of IPV, theories explaining IPV during pregnancy are scarce. The feminist theory, which guided this study, has been used in previous studies to explain IPV in pregnancy [11,57,58]. This model posits that IPV is a result of patriarchal systems where men are the primary perpetrators and women are the primary victims [59,60]. Patriarchy refers to societies where men dominate authority in the family [61]. The feminist theory explains some outcomes of patriarchy, such as gender norms and expectations, economic dependence, and power imbalances. The theory suggests that societies with high levels of gender inequality, including Uganda, are more likely to experience IPV [19,62]. The theory also suggests that violence perpetrated by men in intimate relationships stems from historical and current power imbalances that perpetuate the subordination of women. Men use various types of control (e.g., economic, sexual, physical, and psychological abuse), as well as techniques such as isolation and intimidation to maintain this subordination. Jealousy and possessiveness may also contribute to IPV during pregnancy because the male partner may perceive the unborn child as a rival for the mother's love and attention [63,64]. This theoretical framework provides a foundation for understanding experiences of IPV, support seeking, and coping strategies among pregnant women. Therefore, the question that guided this study was: How do pregnant women describe their IPV experience, help-seeking behaviors, and coping strategies after experiencing IPV?

## Methods

### Study design and setting

This qualitative study employed a phenomenological design to examine how pregnant women experienced, interpreted, sought support, and coped with IPV. This is a qualitative research approach that seeks to explore and comprehend an individual's actual experiences related to a specific event or phenomenon [65]. The study was conducted over the course of one month (January 7, 2019–February 7, 2019) at the ANC clinic of a busy public healthcare institution in Mbarara city.

Mbarara City is located in Mbarara District in Southwestern Uganda. The participating healthcare facility (Health Centre IV) offers ANC, maternity and emergency obstetric care services. Approximately 50–70 expectant mothers from urban and peri-urban areas of the city visit the facility each ANC clinic day. Midwives provide ANC consultations to low-risk expectant mothers, and general practitioners (medical officers) handle cases considered at greater risk. The ANC providers in the study setting had not received any kind of IPV training. However, the ANC card currently used includes a gender-based violence-screening item.

### Study population and recruitment

This study was conducted as part of a larger study which aimed to determine the prevalence of IPV disclosure and associated factors among pregnant women using a sample of 283 [66]. A sample of 283 pregnant women were enrolled and screened for IPV using the WHO study questionnaire [1], which had been used in other sub-Saharan African IPV studies [67,68]. The tool was translated into Runyankore, which is the predominant local language in the study area. Pregnant women aged 15–49 years who had experienced IPV in the index pregnancy were enrolled. Trained research assistants who had completed a Bachelor of Science in Nursing initially screened the pregnant women. Of 283 pregnant women screened, 199 had experienced at least one form of IPV during their index pregnancy.

In the current qualitative study, the lead researcher (midwife) then used purposive sampling techniques to select 25 participants for in-depth interviews among a total of 199 women who had experienced IPV [69]. Selected participants were aged 15–49 years, were pregnant at the time of the study, had experienced IPV during their current pregnancy, and were willing to provide informed consent. Participants were contacted/recruited at the clinic exit after receiving all ANC services.

An interview guide was pre-tested before the in-depth interviews with five pregnant women in an ANC clinic in a nearby public healthcare facility (Mbarara Regional Referral Hospital). Any unclear questions on the guide were corrected (S1 Table). The interview guide included questions about demographic characteristics (e.g., age, ethnicity, education level, occupation, marital status, gravidity, and trimester) and aimed to examine the types of IPV that pregnant women had experienced, whether they sought support, and the coping strategies used. Pregnant women were also asked whether midwives regularly talked about IPV during ANC, and whether they had ever chosen to share their IPV experiences with midwives. The interview guide was developed based on existing IPV literature [41,48,70,71].

The lead researcher and two research assistants conducted in-depth interviews in the participants' native language (Runyakore) in private rooms at the ANC clinic to ensure privacy and prevent interruptions. Written informed consent was obtained before each interview. Pseudonyms were used to ensure participants' anonymity. Interviews were audio-recorded and field notes were taken during the interviews. Each interview lasted 45–60 minutes. Data saturation was reached with 25 participants, which was the point when no new information emerged and responses became repetitive. This confirmed existing patterns and indicated that additional interviews would be unlikely to provide further insights [72]. However, two further interviews were conducted to confirm the study findings.

### Data analysis

The interview data were analyzed using thematic analysis [73], which allowed the meaning of various coded responses from interviews to be analyzed and understood in their proper context [74]. Three researchers (KE, AJB, and NJB)

translated the Runyankore transcriptions of the interviews into English. The researchers then repeatedly read the translated transcripts to familiarize themselves with the data and identify relevant responses related to the study objectives. In each interview transcript, responses containing recurring expressions were identified and color-coded for easy categorization. The color-coded phrases were then arranged into categories (subthemes). Next, categories were deductively related to the aims of the study (themes). New and emerging themes were also inductively identified. The team revised the themes to ensure that they accurately described participants' views and verified their accuracy through follow-up interviews until data saturation was reached. The analysis was completed several years (3 years) after data collection because of the lead researcher's additional educational transitions, which delayed the analysis and dissemination of findings. However, the findings remain pertinent to current debates and deficiencies in the literature about IPV during pregnancy.

## Ethical considerations

The research protocol was reviewed and approved by the Research Ethics Committee of Mbarara University of Science and Technology (Ref: MUREC# 22/09–18). This study adhered to the WHO guidelines on ethical and safety recommendations for research on IPV [75]. The rules and regulations of the Declaration of Helsinki were followed when conducting this study by respecting participants' rights and upholding ethical standards to secure their safety and preserve their health [76]. Informed consent was sought from all participants with assurance that their personal details and information would be kept anonymous. Participants were informed that their participation was voluntary and that their decision to withdraw from the study would not affect their routine care. Parental consent was not required since pregnant women aged 15–17 were considered as emancipated minors. To reduce potential underreporting of IPV experiences, female interviewers conducted interviews in private rooms. For this study, a professional counselor was also enlisted to provide emotional support to women experiencing IPV as necessary. During their interviews, participants received refreshments (soft drinks and snacks). No monetary incentives were offered.

## Rigor

The criteria established by Guba and Lincoln were strictly followed to ensure the credibility, dependability, confirmability, and transferability of the study findings [77]. To maintain the credibility of the results, the research team conducted training meetings and two pilot interviews to enhance their interview skills. The transcribed interviews were examined by the qualitative data analysts for an extended period. In addition, participants validated the data to ensure appropriate representation. Interview summaries and preliminary themes were shared with participants. Participants' feedback confirmed the authenticity of interpretations and helped refine the final analysis.

To ensure dependability, a team of three researchers (KE, NJB, and AJB) independently read and coded the transcribed data, and generated themes from the data. Following this independent analysis, the data were assessed for consistency. Any analyses that were inconsistent or equivocal were discussed among the researchers and interpretations were clarified until consensus was reached. A thorough description of the study approach also allowed for dependability, and permits the study to be replicated. To enhance confirmability, researchers with diverse clinical and training backgrounds conducted the analysis. To promote a common interpretation of the results, each researcher kept a separate reflexive journal in which they documented any problems that may have affected the analysis. Recorded audio files and field notes were used as a form of triangulation to validate the collected data. This minimized the influence of individual researcher's perspectives and increased the rigor and trustworthiness of the research process. This helped ensure the findings were based on participants' responses. The research team conducted additional interviews to confirm the study findings. To enhance transferability, in-depth verbatim quotes were provided to allow readers to judge the application of the present findings to their context.

**PLOS** **Global Public Health**

## Results

In this section, pseudonyms are used and direct quotes are presented as evidence of participants' assertions.

### Participants' characteristics

Table 1 presents participants' characteristics. Most participants (n = 23) were married. The majority were in their second trimester (n = 16), were prime gravida (n = 10), had small businesses (n = 10), and had completed secondary education (n = 14). One participant was abandoned by her partner after discovering they were pregnant, and another ran away after conception because of unbearable violence.

### Main themes identified from the interviews

Four main themes were identified from participants' narratives: power and control dynamics in intimate relationships, drivers of IPV, coping strategies, and barriers to seeking support (S2 Table).

### Theme 1: Power and control dynamics in intimate relationships

The stories shared by participants indicated a range of violent behaviors that included physical, reproductive, psychological, and sexual violence, as well as financial control and manipulation. Pregnant women faced physical violence that involved their gravid abdomen being stepped on and being slapped, pushed, or kicked.

> *Every time I am pregnant, he beats me terribly. He recently stepped on my pregnant belly.* (Phiona)

**Table 1. Participants' characteristics (N = 25).**

| Variable | Frequency (%) |
|---|---|
| **Mean age** (years) | 26.2 |
| **Mean gravidity** (number of pregnancies) | 2.2 |
| **Trimester** | |
| 1st | 3 (12) |
| 2nd | 16 (64) |
| 3rd | 6 (24) |
| **Occupation** | |
| Salaried job | 6 (24) |
| Small-scale business | 10 (40) |
| Housewife | 5 (20) |
| Others | 4 (16) |
| **Education level** | |
| No formal education | 1 (4) |
| Primary | 4 (16) |
| Secondary | 14 (56) |
| Tertiary | 6 (24) |
| **Marital status** | |
| Married | 23 (92) |
| Not married | 2 (8) |

Note: Each trimester is about 12 weeks.

Many participants attributed the physical abuse to their failure to fulfill certain demands set by the perpetrator. For example, one participant shared that before becoming pregnant with her first child, she faced accusations of being unable to conceive and frequently endured severe physical punishment.

*He accused me of not wanting to get pregnant and hit me on the forehead, I had even conceived around that time but both of us had not realized.* (Allen)

Participants experienced IPV for several reasons related to their pregnancy status, including abandonment after disclosing their pregnancy.

*I do not know where he is now; he left me when I told him I was pregnant and blocked my phone number.* (Peace)

Regarding psychological violence, they experienced verbal abuse from their male partners who accused them of extra-marital affairs or became jealous upon seeing them conversing with other men.

*Every time he tells me that I have other men, yet I do not. This makes me feel bad.* (Mable)

Other forms of psychological violence experienced by participants comprised constant verbal abuse from husbands/partners or being forced out of the house because of their crying babies. These actions often caused pregnant women to feel unhappy and depressed.

*He abuses me so much, I just feel so empty inside me, like there is a big hole that cannot be filled.* (Juliet)

Some participants experienced sexual violence, such as being forced to have sex when they did not want it, especially after a quarrel. Others were deliberately forced into uncomfortable positions during sexual activity. These women mentioned that their partners preferred missionary sexual positions and gave little attention to their complaints. This position was extremely painful and uncomfortable because of the gravid uterus in the later stages of pregnancy.

*Laughs…my husband wants to be on top of me during sexual intercourse yet I am pregnant and I feel a lot of pain, I have told him that it is not comfortable, but he does not listen, he does it every day and it hurts me.* (Rose)

*My husband has bad behaviors, I come from the garden very tired, and he forces me to have intercourse with him there and then…* (Allen)

Participating women also described experiencing financial control and manipulation. This included their partners preventing them from working and earning their income while denying them financial support for medical treatment.

*I am a housewife, but when I am sick, he cannot give me money to go for treatment, as if he does not care and it hurts.* (Scovia)

Some participants also mentioned that their partners failed to purchase food for the family, but still anticipated being provided with food upon returning home. Some partners excluded pregnant women from the process of acquiring family assets, leaving the women to learn about these property purchases through friends. Furthermore, some pregnant women who had started up small businesses had their earnings taken by their partners, making it challenging to secure funds for their personal and medical care needs. Such experiences occasionally led to interpersonal conflicts with their partners, which culminated in physical violence.

*He is buying property behind my back. I think it is because I am not financially contributing, but I am worried he might be planning to marry another woman.* (Joyce)

## Theme 2: Drivers of IPV

When asked about the drivers of IPV, participants alluded to several reasons, but most emphasized the existence of a cycle of intergenerational violence in their communities. Several pregnant women reported that their mothers-in-law and biological mothers had experienced similar abuse in the past and often advised them to endure this violence from their partners. This support for intergenerational violence led to the normalization of persistent violence by portraying IPV as unavoidable and socially acceptable.

*This is not new to me because I used to witness my father beating my mother and occasionally kicking her out of the house.* (Brenda)

*Considering the way my spouse's father has treated her in the past, my mother-in-law told me to be patient and know that I will succeed in the end.* (Rehema)

Several participants expressed concern about the unequal power dynamics within households. Men as the sole decision-makers at home often disregarded their wives' contributions. Some pregnant women reported that their partners and other family members conveyed the message that only men are allowed to make decisions regarding their families. As a result, they were prohibited from participating in the decision-making process, which was accepted by the community.

*He makes all of the decisions in our house, and he usually reminds me of it. If I disagree, he will hit me. Many people tell me, "Yes, a man should make decisions in a home," when I ask for guidance.* (Medius)

Participants also highlighted that differences in traditional gender roles and social norms played a role in violence directed at pregnant women. They described strong patriarchal traditions, where men were seen as dominant and the main breadwinners in their families. When they reached out for assistance, some women were advised to endure the abuse.

*My mother-in law always tells me to endure like women of long ago such that I build the home and continue taking care of my children.* (Irene)

Certain women tolerated the violence because they believed that being married conferred status/prestige and social respect. Women were expected to remain obedient, focus on home and childcare, and some refrained from earning an income at the request of their partners. Participants noted that to assert control, their partners claimed ownership of household assets. Such relationships heightened the risk for IPV.

*My husband asserts that since he pays the school fees and constructed the house, I should undertake the donkeywork, even when I am unwell, he refuses to assist me.* (Annette)

## Theme 3: Coping strategies

When asked how they dealt with their IPV experiences during pregnancy, participating women voiced several coping strategies. These were categorized as 1) active coping strategies and 2) accommodating abuse (passive coping strategies). Active coping strategies involved women controlling their emotions and managing challenges. Pregnant women

approached and asked their friends, and relatives to support them, turned to positive thinking, and prayed for IPV to end. These strategies helped participants regain their sense of control, reduced stress, and helped them work toward safety.

*Almost every day I call my mother such that we talk about it and I relieve the stress off me. My mother tells me to keep quiet and ignore him.* (Sandra)

*…After injuring me I reported to the chairman and police, he was counseled and now he is better.* (Allen)

Others sought support from church leaders such as pastors, who counseled and prayed with them.

*I always pray to God and my pastor (church leader) prays for me; I have faith that he will eventually change.* (Sonia)

Some participants coped through accommodating the abuse, such as agreeing to everything their partner said.

*Each time he says something whether wrong or right, I say yes yes…to avoid being beaten.* (Marion)

Others tolerated their partners' behaviors, including being beaten, without opposition and chose to remain silent during violence. Some managed to avoid confrontation by being aware of their partner's preferences.

*Most of the time he quarrels, and I pretend to be a fool, so I always keep quiet to avoid slaps and punches.* (Rita)

*…Everyone has their battles; I choose to accept my husband despite his shortcomings. Many families are in a similar dilemma.* (Rehema)

**Theme 4: Barriers to seeking support**

Some participants sought help after IPV as an active coping strategy, whereas others did not. The latter were questioned further to understand the obstacles that prevented them from seeking assistance. They identified several barriers: pregnant women's lack of awareness of the supportive role of healthcare workers (midwives) in IPV management, caregiving challenges, shame and guilt, fear, partner dependence, and community endorsement of the abuse. Participants thought that midwives had no role in matters concerning their private family affairs.

*I do not think a midwife has a role to play in this abuse; is she my aunt?* (Irene)

Participants voiced that they had trouble-approaching midwives and this made it hard to disclose their IPV issues.

*I want to tell the midwife about my problems at home, but I do not know how to start.* (Rose)

When these participants were further questioned on whether healthcare personnel inquired about IPV, several stated that they were never asked about IPV and that healthcare workers' primary focus was on abdominal examinations and they ignored pregnant women's psychological needs. Pregnant women further noted that the limited number of staff on duty translated into less time available for each client. This made it difficult for these professionals to routinely screen for IPV.

*No one has ever asked me; they are primarily interested in examining the abdomen.* (Rose)

Instances of community endorsement of abuse were observed, as some participants reported receiving no support after disclosing experiences of IPV to friends. This lack of support served as a deterrent to seeking further assistance

from other sources. Some participants found solace in their conviction that their abusive partners still loved them, whereas others indicated that they would remain in their abusive relationship as long as their partner continued to provide food, financial support, and conjugal rights.

*If he is still interested in having sexual relations with me and pays fees for the children, it shows he still loves me so I cannot think of separating from him. I will bear his abuses.* (Generous)

In addition, some participants mentioned that requesting assistance was challenging, particularly following an injury. They were afraid of being mocked or embarrassed. Some never sought help because they had fled a previous violent relationship and were consequently worried about being harshly judged by society for leaving a second relationship. Others perceived IPV as a private family issue that, at the very least, should remain confidential and not be disclosed to outsiders or those not involved.

*Those are bedroom issues that should not be discussed outside and if you tell anybody, they will move around telling others about what happened to you and I find that is humiliating.* (Annette)

Some pregnant women wished to protect their violent partner's public image.

*I lied to healthcare workers about the injuries I sustained during the fight. I felt it was embarrassing to disclose.* (Janet)

Furthermore, participants did not seek assistance because they were fearful of experiencing retaliation from abusive partners if they became aware of the assistance requested.

*I told my brother-in-law about the abuse, but it made things worse. My husband attacked me that night...since then I fear telling anyone.* (Sarah)

Some participants reported that their abusive partner threatened to abandon them and their children if they reported the abuse. As a result, they chose to keep the abuse to themselves.

*Every time he beats me, he threatens that he will abandon me with the children and never come back if I report to any-one, I cannot afford the finances to care for my children so when I hear that I cool and leave everything.* (Brenda)

## Discussion

This study found that pregnant women experienced various forms of IPV including physical, psychological, and sexual violence, and financial control and manipulation. This was consistent with previous studies conducted in Uganda, Kenya, Tanzania, Ethiopia, South Africa, and low-income communities in the US [2,6,17,23,26,71,78,79]. The consistency of these forms of IPV across these diverse regions suggests that these types of abuse are common irrespective of the cultural, economic, or geographical context [11]. This study highlighted that IPV can manifest in different ways across various contexts. Healthcare systems should guarantee the implementation of routine IPV screening during ANC and enable fast referral services, thereby facilitating early identification and support for affected women.

In violent partnerships, physical abuse remains a common form of violence. Research suggests that pregnant women are vulnerable to IPV, and some abusers use pregnancy as an excuse to commit violent acts against women [11]. Some participating pregnant women reported their gravid bellies being kicked and stepped on. This was consistent with a previous study conducted in Nicaragua where pregnant women were reported to have been punched and kicked in their

abdomen [80]. IPV perpetrators purposely targeted the abdomen to inflict discomfort, anxiety, and possible injury on the fetus as well as the mother [81,82]. Intentional violent abdominal hitting during pregnancy is a serious concern that impacts the fetus and the mother's overall health [82]. As a result, this study recommends targeted education for men to address the root causes of hostility toward pregnant women and the unborn and help them find alternative outlets for their emotions than violence.

In this study, pregnant women narrated how their partners deliberately insisted on uncomfortable sexual positions, such as the missionary position. This was consistent with previous studies conducted in Nigeria, Brazil, Poland, Portugal, and the US, which reported that the missionary position remained common during pregnancy [83–87]. Cultural standards around masculinity can result in the exercise of power and control in intimate relationships, particularly through abuse and manipulative actions such as insisting on specific sexual practices. This intentional abuse can be linked to these norms [88]. It has been demonstrated that IPV decreases when intervention programs involve males in discussions about respectful partnerships, shared decision-making, and the risks for IPV [89]. Promoting male participation in prenatal care can help healthcare workers cultivate favorable attitudes toward shifting from harmful cultural attitudes and reducing IPV. For effective communication during sensitization, midwives or other healthcare providers could use internet resources and other instructional materials to reinforce the message.

Pregnant women in this study voiced concerns about financial control and manipulation by their partners in various ways, including financial secrecy and prohibiting them from working for payment. This finding was consistent with previous studies from Ghana, Vietnam, and New Zealand [70,90,91]. These findings highlighted the significance of addressing women's empowerment challenges as a way to improve their well-being [51,92]. Therefore, key stakeholders such as healthcare providers, human rights activists, legislators, and parents need to engage with men regarding the acceptability of women's empowerment. Furthermore, it is important to work toward empowering girls and women through better education, reskilling, up-skilling programs, and problem-solving skills training. This will create more opportunities for income generation, enhance their ability to handle various financial challenges, and prevent women from being overly dependent on men. Previous studies have consistently showed that economic dependence was the main obstacle that victims faced when trying to leave violent relationships [3,4,93]. Women's economic dependence on their partners makes them more vulnerable to sustained abuse [94].

In this study, pregnant women reported intergenerational violence as a driver of IPV, which was consistent with prior research conducted in sub-Saharan African countries [95–98]. Children who witness IPV experiences are more likely to replicate such behaviors in their own adult relationships [99,100]. To break the cycle of violence, midwives and other stakeholders should use prenatal care and outreach as opportunities to educate and sensitize communities about the dangers of IPV and promote positive, non-violent relationships. In addition, implementing programs such as stress management classes, parenting education and anger management can help families that may be experiencing or engaging in violent behaviors. We also advocate for the implementation of community-level interventions to prevent and mitigate IPV among pregnant women, as recommended in a study from Uganda that showed community mobilization can greatly decrease IPV [101]. Such approaches should be integrated into maternal healthcare initiatives.

This study showed that unequal power dynamics and gender norms served as catalysts for IPV. Men made most decisions in their homes without considering women's concerns. This often led to IPV and was normalized by participants' communities. This was consistent with the feminist theory, which holds that social and cultural contexts play a role in how women experience IPV. The model also promotes female autonomy and empowerment [57,59]. Arguably, the notion of men traditionally restricting women from active decision-making emboldens them to maintain control and power over women and ultimately perpetuates violence [102,103]. The feminist theory emphasizes changing social norms and attitudes to prevent violence against women by increasing public awareness and ultimately changing behaviors. Therefore, policy efforts should focus on empowering women by enhancing their self-esteem, ensuring they have control over their lives and resources, and promoting shared decision-making at the household level to combat harmful gender norms.

This study found that pregnant women used active coping mechanisms to minimize the negative effects of IPV, such as sharing their IPV experiences with friends, family members, and church leaders (pastors) for support. This concurred with previous studies that were conducted in Tanzania, Nigeria, Iran, and Bangladesh. Women from the aforementioned countries used proactive strategic responses such as seeking help from others to overcome the negative effects of experiencing IPV [50,51,104,105]. This could be explained by the fact that in many cultures, social support networks (e.g., friends and family) are the first line source of support for anyone in need of counseling and comforting, including pregnant women [104,106,107]. Having someone to confide in is vital, as a lack of victim support acts as a barrier to ending IPV [108]. However, the present findings indicated that the individuals, to whom women confided their IPV problems, were often not supportive, actively encouraged them to tolerate abuse and keep calm to avoid conflict, and discouraged them from sharing family matters with those not directly concerned. This normalized the abuse and discouraged further help-seeking, as reported in previous studies conducted in Uganda, Tanzania, Ethiopia, and the US [109–113]. Therefore, educating community members, particularly those in advisory or support roles, to identify abuse, respond appropriately, and promote help-seeking behaviors instead of normalizing or minimizing abuse is crucial. This study also highlighted the need for additional interventional research into effective social support networks and interventions for pregnant women who have suffered IPV in low-income settings.

Consistent with previous studies from Iran and Bangladesh [50,51], pregnant women in this study employed accommodative coping strategies such as agreeing to their partner's demands and remaining silent about their IPV experiences. This could be partly explained by the fact that traditional gender roles promote passive coping strategies as women are expected to be obedient to their male partners in many African nations, including Uganda [114–116]. Passive coping methods may reduce physical violence in the short term and provide brief stress relief. However, they do not address the underlying cause of IPV, thereby leading to a vicious cycle of violence. Further research is needed to determine the long-term impacts of accommodative coping strategies on pregnant women and their newborns when compared with those who employ active coping strategies in Sub-Saharan Africa.

A key barrier to support seeking reported by pregnant women in this study was women's lack of awareness about midwives' role in IPV management, which was consistent with a study conducted in Jordan [117]. This could be attributed to pregnant women's limited knowledge about midwives' roles in IPV care. In many communities, the role of midwives in resolving violence against women is sometimes misunderstood. Most community members typically view the role of midwives as restricted to childbirth. Therefore, women may not consider midwives a possible source of help and support, especially when dealing with IPV [117]. This calls for maternity and child health policymakers to develop and implement public awareness campaigns to educate communities (especially women) about the role of midwives in the care and prevention of IPV and its consequences, and encourage women to seek midwives' support. In this study, the lack of knowledge of midwives' role in IPV among pregnant women was exacerbated by midwives' failure to inquire about IPV. Similarly, a study conducted in Norway found some midwives did not routinely inquire about IPV [118]. This could be because midwives lack sufficient time, knowledge, or training in recognizing and dealing with IPV, which would result in a lack of experience or confidence when discussing IPV with women [117,119,120]. Other reasons midwives may not inquire about IPV include their own patriarchal norms and personal experiences of IPV victimization [51,118,121]. To ensure practicing midwives remain up-to-date with the most recent evidence-based approaches and strategies, policy initiatives should seek to provide regular continuing education on IPV through conferences, workshops, and online training courses. Specifically, obstetricians and midwives need expert training on how to recognize and handle IPV to enable them offer routine IPV screening, care, referral, and follow-up.

Several pregnant women in this study expressed shame and guilt as impediments to support-seeking, and others covered up the cause of their IPV injuries, consistent with previous studies conducted in Tanzania, Ghana, Peru, and the US [122–125]. The reason for these findings may be that women who have been victims of IPV may experience embarrassment and not seek support for fear of being stigmatized, held accountable for their situations, and judged [111,126,127].

To address damaging cultural and societal practices that stigmatize or blame women who have experienced IPV, governmental and non-governmental stakeholders should engage cultural and community leaders to develop and implement public awareness campaigns against IPV, promote positive gender roles, and emphasize the need for equality, respect, and nonviolent relationships.

This study had strengths and weaknesses that may influence its conclusions. First, phenomenological qualitative research methodologies were employed, which are considered effective for understanding pregnant women's experiences and coping mechanisms. Second, although previous research on IPV in Uganda concentrated on psychological, physical, and sexual abuse, this study additionally considered the economic aspect of violence. However, data were collected in 2019, which means that there may be contextual differences between then and now. A major strength of this study was that it covered an area of Uganda that was underrepresented in previous studies. The study having been conducted in one clinic may have introduced bias. In addition the fact that most themes were deductively linked to the study aim may have introduced researcher bias in coding.

## Conclusion

This study revealed that pregnant women experience various forms of IPV. The main drivers of IPV among pregnant women were intergenerational violence, feelings of helplessness, unbalanced power dynamics, and gender roles. Although some pregnant women tried to deal with IPV using active coping strategies, others used passive coping strategies. Several barriers, including a lack of awareness about midwives' role in IPV management, shame, guilt, and fear of reprisal, meant some women tended not to seek any support. The findings indicate that stakeholders need to prioritize IPV screening, care, referral, and sensitization in healthcare facilities and communities. In addition, rules and regulations that protect the rights of IPV survivors should be strengthened, and perpetrators held accountable for their actions. There is also a need to learn from IPV implementation studies in other populations to develop and evaluate the effectiveness of various interventions that prevent and support the victims of IPV during pregnancy.

## Supporting information

**S1 Table. Interview guide.**
(DOCX)

**S2 Table. Themes that emerged from participants' responses.**
(DOCX)

## Acknowledgments

The authors express their gratitude to the First Mile Community Health program for their support during this study. We also extend our gratitude to the Department of Community Health at Mbarara University of Science and Technology for providing support. Finally, we thank the study participants for their willingness and cooperation throughout the data collection process.

## Author contributions

**Conceptualization:** Eve Katushabe, John Bosco Ndinawe, Editor Abeneitwe, Agnes Katusiime, John Baptist Asiimwe, Vincent Batwala.

**Data curation:** Eve Katushabe, John Bosco Ndinawe, John Baptist Asiimwe, Vincent Batwala.

**Formal analysis:** Eve Katushabe, John Bosco Ndinawe, John Baptist Asiimwe.

**Funding acquisition:** Eve Katushabe.

**Investigation:** Eve Katushabe, Editor Abeneitwe, Agnes Katusiime, John Baptist Asiimwe.

**Methodology:** Eve Katushabe, John Bosco Ndinawe, Editor Abeneitwe, John Baptist Asiimwe, Vincent Batwala.

**Project administration:** Eve Katushabe, Vincent Batwala.

**Resources:** Eve Katushabe.

**Supervision:** Eve Katushabe, Vincent Batwala.

**Validation:** Eve Katushabe, John Bosco Ndinawe, Vincent Batwala.

**Visualization:** Eve Katushabe, John Baptist Asiimwe, Vincent Batwala.

**Writing – original draft:** Eve Katushabe, John Bosco Ndinawe, Editor Abeneitwe, Agnes Katusiime, Gladys Nakidde, John Baptist Asiimwe, Vincent Batwala.

**Writing – review & editing:** Eve Katushabe, John Bosco Ndinawe, Editor Abeneitwe, Agnes Katusiime, Gladys Nakidde, John Baptist Asiimwe, Vincent Batwala.

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
