## [Decision Letter · Decision Letter 0]

27 Feb 2024

PGPH-D-23-02096

“He stepped on my belly” An exploration of Intimate Partner Violence Experience and Coping Strategies among Pregnant Women in Southwestern-Uganda

Dear Dr. Katushabe,

Thank you for submitting your manuscript to PLOS Global Public Health. After careful consideration, we feel that it has merit but does not fully meet PLOS Global Public Health’s publication criteria as it currently stands. Therefore, we invite you to submit a revised version of the manuscript that addresses the points raised during the review process.

Your manuscript has been evaluated by two reviewers, and their comments are appended below.

The reviewers have provided a range of comments to address, in particular regarding the study design and description of the methodology, and the overall clarity of communication. Please address in particular the comments from Reviewer 1 regarding the discussion section, and ensure that all statements are fully supported by data in this study or appropriate literature references. Please ensure you address each of the reviewers' comments when revising your manuscript.

We look forward to receiving your revised manuscript.

Kind regards,

Hugh Cowley

Staff Editor

Journal Requirements:

1. Please provide copy editing.

Additional Editor Comments (if provided):

Reviewers' comments:

Reviewer's Responses to Questions

**Comments to the Author**

1. Does this manuscript meet PLOS Global Public Health’s publication criteria?

Reviewer #1: Yes

Reviewer #2: Yes

2. Has the statistical analysis been performed appropriately and rigorously?

Reviewer #1: N/A

Reviewer #2: Yes

3. Have the authors made all data underlying the findings in their manuscript fully available (please refer to the Data Availability Statement at the start of the manuscript PDF file)?

Reviewer #1: Yes

Reviewer #2: Yes

4. Is the manuscript presented in an intelligible fashion and written in standard English?

Reviewer #1: No

Reviewer #2: No

Reviewer #1: Important issue – very high prevalence of IPV during pregnancy in East African countries, including Uganda with many physical and mental health implications for women, and physical health implications for babies.

Overall: Identifying and applying a theoretical framework would strengthen the analysis and discussion. Considerable language and grammar editing needed, including in quotations in results section, and in discussion section. Many distinct sentences combined into one sentence, quotation marks missing, capitalization in the middle of sentences, bold font used where inappropriate, etc. Formatting of headings/subheadings needs to be improved so that the level of subheading is clear.

Lines 80-81 – Some editing required. “Nevertheless,”. Can you be more specific re. “a number of victims” and clarify how this sentence relates to previous one, “yet involving other people can be a strategy to cope with the violence.”?

Lines 85-88 – Some additional commas needed in list of coping strategies.

p. 4-5. Can you clarify distinction between separate lists of coping strategies, or synthesize them?

Line 98. Use full word for ANC before using acronym.

Line 100 – What do you mean by “ANC package”?

Line 118 – revise to “2018 hospital database records of new and returning ANC cases.”

Lines 118-122 – some editing needed to add commas and break it down into 2 sentences.

Line 128 – revise to “in-depth”

Lines 130-138 – editing required to more clearly and grammatically appropriately outline interview guide questions, or outline topics included in interview guide.

Lines 139-140 – Editing required, and more description of field note purpose and process.

Line 158 – Revise “scripts” to “transcripts”.

Line 185 – As this section describes methods, don’t include how many women sought emotional support here.

Lines 206-207 – separate into 2 sentences.

Lines 209-213 – Needs further clarification and stronger alignment with intent of Guba and Lincoln’s criteria. Goal to “achieve neutral findings” unclear, and process by which field notes lead to that is also unclear. What do you mean by “nominated sample” and how does giving an adequate description of the study site and respondents help to achieve that?

Lines 219-224 – needs editing

Table 1 – Variable and category columns can be collapsed

Line 232 – Revise to “IPV experiences” to align with themes outlined.

Lines 236-239 – Revise to distinguish between actions of abuser vs. actions of women experiencing IPV – currently unclear.

Line 252 – add “

Results

- Some quotations repetitive of what is in text. Include quotations that support and add to what is synthesized in text.

Discussion

- When connecting findings to previous research, clarify where that research was conducted, and whether those contexts are relevant to where study was conducted. State more about previous research referenced. “The results are similar to previous studies” is too vague.

- Line 663: “women are never breadwinners” – This is not true.

- Line 664: “African tradition” is a very broad generalization for a continent with many, many different countries and communities within those countries.

- Lines 666-667 – Inappropriately suggests that women are responsible for divorce.

- recommendations stated too strongly, without reference to implementation research on those recommendations

Reviewer #2: I really enjoyed reading the paper. I believe it is an interesting research. Here are some comments:

Please do not use contract forms (such as e.g.)

The article's tone lacks academic formality and appears overly informal. I recommend adopting a more scholarly and formal style.

There are several grammatical mistakes and typos in the paper. Please correct them.

Introduction:

The introduction is well-constructed. I suggest improving the references in your paper by incorporating recent publications focusing on interviews with pregnant women. Adding these articles will enhance the depth of your research.

The objective is vague. I suggest you rehrasing it in a more clear way.

Method:

LINES 126-127: Can you provide more information on data saturation? How did you get that 25 was enough?

LINE 154: Please provide a reference for the thematic data analysis approach you used.

PLease provide information on whether you used a software for data analysis or not.

Results:

The results are very interesting but this section is very long. I suggest you to shorten it to make the article more easy readable.

“Theme one: Various experiences by pregnant women” I suggest you rephrase it to “Theme one: Various violent experiences by pregnant women”

491: there is a typo

LINES 513 – 527: too many quotes here. Just pick three.

LINE 527: Please comment this theme as well “Saying yes to everything”

**Do you want your identity to be public for this peer review?** For information about this choice, including consent withdrawal, please see our Privacy Policy

Reviewer #1: **Yes: ** Dr. Farah N Mawani

Reviewer #2: No

---

## [Decision Letter · Decision Letter 1]

18 Jul 2024

PGPH-D-23-02096R1

“He stepped on my belly” An exploration of Intimate Partner Violence Experience and Coping Strategies among Pregnant Women in Southwestern-Uganda

Dear Dr. Katushabe,

Thank you for submitting your manuscript to PLOS Global Public Health. After careful consideration, we feel that it has merit but does not fully meet PLOS Global Public Health’s publication criteria as it currently stands. Therefore, we invite you to submit a revised version of the manuscript that addresses the points raised during the review process.

The revised manuscript has been reviewed and the reviewer comments are available below. Additional suggestions have been made to improve the manuscript such as reporting the inclusion criteria for participants. Please review the comments and revise the manuscript accordingly. 

We look forward to receiving your revised manuscript.

Kind regards,

Emma Campbell, Ph.D

Staff Editor

Journal Requirements:

Reviewers' comments:

Reviewer's Responses to Questions

**Comments to the Author**

Reviewer #3: (No Response)

publication criteria?

Reviewer #3: (No Response)

3. Has the statistical analysis been performed appropriately and rigorously?

Reviewer #3: (No Response)

4. Have the authors made all data underlying the findings in their manuscript fully available (please refer to the Data Availability Statement at the start of the manuscript PDF file)?

Reviewer #3: (No Response)

5. Is the manuscript presented in an intelligible fashion and written in standard English?

Reviewer #3: (No Response)

Reviewer #3: This manuscript shows promise for publication but has some major flaws at this point.

As a qualitative piece it does not yet add new material other than a description of the nature of abuse experienced by this group of women.

METHODOLOGY - this section needs expansion. Include: inclusion criteria for participants, how they were identified as having experienced IPV, where and by who they were approached to participate in the study. Were women paid? Where were interviews conducted?

Line 180 – consent waived for minors. Should this refer to parental consent?

Line 181 - suggest this should read “ identifying information” rather than “identity”

If interpretive phenomenological analysis was used, some rationale for this choice and brief summary of the approach with references should be included.

Rigor Line 196-197 These features guaranteed that data collection and content were consistent. It is suggested that to claim these measures provided a “guarantee” is an unwise. Rather, I suggest you state the measures “ supported” consistency.

FINDINGS:

To establish the credibility of the findings, it is suggested that each theme is defined and articulated. As part of this, and the analysis indicates the strength of the theme and any groups who this did not apply to. As written in the current manuscript, each theme simply includes individual accounts to illustrate the theme. These work as examples of a theme, but do not by themselves constitute complete analysis.

There is no Discussion section and the Limitations are not relevant for qualitative research

Good luck with your endeavours

**Do you want your identity to be public for this peer review?** For information about this choice, including consent withdrawal, please see our Privacy Policy

Reviewer #3: No

---

## [Decision Letter · Decision Letter 2]

7 Jan 2025

PGPH-D-23-02096R2

Intimate Partner Violence Experience, support seeking and Coping Strategies among Pregnant Women in Southwestern Uganda

Dear Dr. Katushabe,

Thank you for submitting your manuscript to PLOS Global Public Health. After careful consideration, we feel that it has merit but does not fully meet PLOS Global Public Health’s publication criteria as it currently stands. Therefore, we invite you to submit a revised version of the manuscript that addresses the points raised during the review process.

We have received two reviews with some detailed feedback below. Please revise to address these comments and suggestions regarding phrasing and reporting.

We look forward to receiving your revised manuscript.

Kind regards,

Julia Robinson

Staff Editor

Additional Editor Comments (if provided):

Reviewers' comments:

Reviewer's Responses to Questions

**Comments to the Author**

Reviewer #4: (No Response)

Reviewer #5: All comments have been addressed

publication criteria?

Reviewer #4: Partly

Reviewer #5: Yes

3. Has the statistical analysis been performed appropriately and rigorously?

Reviewer #4: N/A

Reviewer #5: Yes

4. Have the authors made all data underlying the findings in their manuscript fully available (please refer to the Data Availability Statement at the start of the manuscript PDF file)?

Reviewer #4: No

Reviewer #5: Yes

5. Is the manuscript presented in an intelligible fashion and written in standard English?

Reviewer #4: Yes

Reviewer #5: Yes

Reviewer #4: Thank you for the opportunity to read this interesting manuscript on an important subject. Please note that I did not peer review previous versions and the version provided to me does not have changes tracked.

The analysis of findings requires some finessing. The Discussion, especially, is long and does not always position the findings in relation to the literature. Throughout, the text veers into quantitative or positivist language out of keeping with the qualitative methods used. I outline specific suggestions below.

TITLE

Please note that the title in the manuscript differs from the title in the electronic system.

ABSTRACT

This also differs substantially between the manuscript and the electronic system and I was not sure which version was more recent.

INTRODUCTION

The authors may wish to define intimate partner violence at the outset.

The authors alternate between IPV and the full words and I suggest that IPV can be used throughout after the first definition of the acronym.

Line 53 - I am confused about the prevalence of IPV being 27% at all of the global, regional and national levels. I would expect prevalence estimates at each of these levels would differ somewhat.

Line 57 - I understand what the authors mean by "IPV is a tradition" - that there is a long history of it going back generations - but the phrasing could potentially be interpreted as justifying IPV as a cultural practice so I suggest rephrasing this sentence.

Line 60 - the prevalence is stated above to be 40% so it does not seem accurate that "most" women experience IPV.

The authors may wish to acknowledge that many of the coping strategies identified focus on the woman accommodating abuse rather than addressing male behaviour.

Line 98 - how does IPV screening effectively reduce depression?

Line 100 - "women's qualitative inquiry of IPV experiences" - do the authors mean women's qualitative accounts of IPV or staff members' inquiry about IPV?

Line 122 - would feminist theory say anything about pregnancy being a time of IPV vulnerability because the unborn represents a rival to the man in a patriarchal society?

As a qualitative study it would help to explicitly state the research questions the authors set out to answer.

METHODS

Line 146 - please explain how saturation was defined and determined.

Line 155 - "the potential reasons behind it" sounds as though IPV is sometimes "reasonable" or justifiable. Perhaps this could be rephrased eg "associated factors" or "contextual factors".

Line 173 - the authors say earlier that they used an interpretive phenomenological approach but then say they used thematic analysis. Can they provide more detail about their analysis and how IPA came into it?

Line 176 - what does "checked themselves" mean?

Line 178 - the term "significant" in relation to responses sounds rather quantitative. Could the authors clarify what they mean?

Line 180 - "repetitive" sounds a bit critical. Recurring?

Line 185 - I think this should say "responses" rather than "respondents". The term "accuracy" is rather quantitative-sounding.

Line 198 - participants aged 15-17 years are likely to have been the most vulnerable. Can the authors elaborate on the concept of emancipated minors and why they would not need to consent?

Line 203 - I assume refreshments were for women's comfort and well-being? Was there a meaningful risk of hypoglycemia?

Line 210 - again, "perfect" is a rather positivist term. Enhance interview skills?

Line 212 - Data is plural of datum so should be written as "data were" rather than "was" in all cases.

Line 215 - "for dependability the study to dispose of individual analyst predisposition" - this is unclear - I think it is about attaining consensus and confronting bias but this needs to be more clearly explained.

Line 223 - I think a semi-colon is needed before "this contributed to appreciation..."

Line 226 - removing bias and producing objective results are quantitative rather than qualitative objectives. It would help to consistently frame the methods in qualitative terms of rigour and credibility rather than objective truth.

Line 228 - transferability is not something that one can "guarantee". Again I suggest less positivist language.

There is no mention of assistance provided to women for the abuse they disclosed, or safety protocols to address risks identified during participation.

RESULTS

Line 251 - this sentence has some formatting errors: a capital P in the middle of the sentence, a space before the comma.

Line 257 - the statement about feeling depressed is not supported by the quotations below. This (impact on mental health) is an important point so I suggest providing a supportive quotation.

The section on IPV is broken down into straightforward categories but some of the quotations implicate manipulative behaviour that shows signs of coercive control; I wonder if the authors wish to acknowledge this.

Line 325 - "claiming to have no biological mothers" sounds like the women are not being believed. Could this be rephrased?

The groupings of types of coping are rather simplistic. These could be less granular and more thematic, for example grouping all the ways that women adapt their own behaviour to accommodate abuse or all the ways members of the community inadvertently reinforce abuse.

Women in quotations are often laughing ruefully but humour isn't mentioned as a coping strategy.

The fractured family unit, for example missing parents (I imagine, due to HIV, although this is not mentioned) is not mentioned. This could be developed further within the intergenerational transmission theme which is currently very brief (and might also fit better as a sub-thene of experiences rather than a stand alone theme).

Overall the analysis could be developed to be more inductive and bring together more thematic understanding.

DISCUSSION

Line 450 - there are some grammatical errors in this sentence - an apostrophe after midwives and capital A used for antenatal care.

Line 457 - these types of violence are reported worldwide, not only in LMICs.

Line 470 - "prevents most women from working hard". This sounds rather judgemental and I would suggest a more nuanced discussion of economic gender dynamics.

Line 474 - this discussion of the need for economic empowerment does not draw on literature on this topic or the potential for increased violence if patriarchal norms are threatened. Economic Empowerment is clearly important but considerations of potential unintended harms is essential.

Line 483 - New Zealand?

Line 491 - which proof is being alluded to here?

Line 511 - while the authors are to be commended for discussing safe sexual positions freely as well as women's comfort and pleasure, the length of description feels excessive and does not obviously relate to the focus of the paper.

Line 535 - where was this study conducted? How can the authors reconcile their findings with those?

Line 539 - again, this section feels excessively long and it is not clearly linking the authors' findings to existing evidence. They seem to be describing best practice but I was unclear whether they were saying what should happen or what does happen in some places.

Line 558 - the authors here again repeat their findings, which have already been stated at the start of the Discussion.

The comments about harms of spending time in church and lack of balance in life neglect the greater concern that religious coping treats violence as an intractable problem and also places the burden of coping on women while taking no action against perpetrators.

CONCLUSION

Line 615 - "ignorant" sounds critical. Not informed?

While reasonable recommendations, few of the conclusions follow from the paper's findings.

Recommending education on safe sexual positions presumes that male partners are unaware of risks to the woman and foetus whereas in this paper, unsafe positions were an example given of deliberate abuse.

Line 643 - qualitative research prioritises lived experience and realities; "recall bias and inaccuracies" do not make sense as limitations of the methods used.

Reviewer #5: I wish to thank the authors for their immense work on generating evidence on such an important topic such as IPV, and more so among pregnant women.

The authors have done a good job responding to all of the reviewer comments. However, they will need to work on language (punctuation and sentence length). The best option would be to submit the work for thorough English language editing. Below are a few suggestions for revision:

Page 4 line 114-117: the sentence is too long and not easy to understand. The first part makes sense, but the confusion starts with the second part and more confusing with the third part. Kindly review for conciseness.

Page 6 line 162-170: It would be great to let the reader know if this facility is public or private run. Also, an idea about its location or catchment area – does this serve high, low or mixed SES groups?

Page 7 line 195-198: The sentence is too long and needs to be broken into at least two for better understanding. It is not clear what the authors seek to communicate – the experience of the research team member? The persons who conducted the interviews? That steps were made to establish rapport ahead of interviews?

Page 7 lines 204-206: this contains 4 related short sentences which can be combined to form two sentences. Fir instance, “The study used pseudonyms instead of data identifiers to ensure anonymity” or “Data saturation was achieved by the 25th interview but we added two more interviews for confirmation.”

Page 8 lines 229-231: The sentence ought to be revised to eliminate a technical fault. Consent is obtained from adults and assent obtained from minors. Thus, @....the need for assent by parents…” is technically unacceptable and must be corrected to “consent by parents”

Page 9 lines 254-256: The sentence needs to be revised for clarity and conciseness

Generally, the authors must use punctuation to make it easier to read some of the respondent quotations used. I do understand that these were verbatim from local dialect to English. However, non-native speakers will find it difficult to understand the translated versions without careful use of punctuations.

Page 14 lines 391-392: revise sentence for clarity. Perhaps insert “and” between IPV management and thought.

Lines396-397: Revise sentence for clarity, The current use of “lack of staff and inadequate time” can be improved. Suggestion, “participants noted that the number of staff on duty often meant less time available for each client, making it difficult for them to routinely screen for IPV”

Lines 398-399: might be better to start a whole new sentence with “some participants indicated….”

Page 17 lines 481-485: The need to empower women economically is a general concept and not limited to the pregnant women in this study. Please revise to eliminate “pregnant” and pregnancy is just transient. Same for sentence in 504-505. Also the last sentence is not complete and appears to be missing some part.

Page 18 line 498-500: Please revise for clarity and conciseness

**Do you want your identity to be public for this peer review?** For information about this choice, including consent withdrawal, please see our Privacy Policy

Reviewer #4: **Yes: ** Dr Roxanne Keynejad

Reviewer #5: No

---

## [Decision Letter · Decision Letter 3]

29 Apr 2025

PGPH-D-23-02096R3

Intimate Partner Violence Experience, support seeking and Coping Strategies among Pregnant Women in Southwestern Uganda

Dear Dr. Katushabe,

Thank you for submitting your manuscript to PLOS Global Public Health. After careful consideration, we feel that it has merit but does not fully meet PLOS Global Public Health’s publication criteria as it currently stands. Therefore, we invite you to submit a revised version of the manuscript that addresses the points raised during the review process.

We look forward to receiving your revised manuscript.

Kind regards,

Miquel Vall-llosera Camps, Ph.D.

Staff Editor

Journal Requirements:

1. We would like to request a copyediting.

Additional Editor Comments:

The previous reviewer has raised remaining concerns that need to be addressed.

Reviewers' comments:

Reviewer's Responses to Questions

**Comments to the Author**

Reviewer #4: (No Response)

publication criteria?

Reviewer #4: Yes

3. Has the statistical analysis been performed appropriately and rigorously?

Reviewer #4: N/A

4. Have the authors made all data underlying the findings in their manuscript fully available (please refer to the Data Availability Statement at the start of the manuscript PDF file)?

Reviewer #4: No

5. Is the manuscript presented in an intelligible fashion and written in standard English?

Reviewer #4: Yes

Reviewer #4:  The paper would benefit from editing, primarily to add coherence to the results, providing clearer grouping among the themes and more interpretive analysis. There is some repetition which could be removed to make the paper more concise. I outline specific points below.

ABSTRACT

Coercion is used interchangeably with violence here but it is a subtype of violence and abuse. E.g. I would not call violation in pregnancy "reproductive coercion" - I would call it rape or sexual violence.

Some phrases are made less clear by being truncated, eg "women's caregiving perceived challenges" - can these be rephrased to be clearer?

INTRODUCTION

Line 68 "several women" seems a bit misleading given the high prevalence mentioned in the preceding sentence.

Line 72 "is a tradition" sounds a bit like cultural normalisation. Alternative wording would be helpful.

Line 73 "vice" is also an unusual term here. Break the cycle, perhaps?

Line 74 "above all some of the women still believe..." this could be interpreted as victim blaming. Perhaps it's more accurate to say that women continue to be socialised to accept unequal rights and roles?

Line 76 "linked several factors" sounds like a causal interpretation. Perhaps "identified several risk factors" for IPV is more accurate.

Line 107 alexithymia is unexpected here. Can the authors clarify why women would experience this due to IPV?

Line 107-112 care is needed when describing ways that women accommodate IPV by changing themselves to placate abusive partners. I suggest some adjustment to the wording here to clarify that studies have identified a range of ways that women may seek consolation, support or safety (rather than it sounding like these are all good ideas).

Line 125-127 - do these points (8x ANC contacts and IPV screening) refer specifically to Uganda?

Line 128 in what way does IPV screening "effectively reduce depression"?

METHODS

It wouldhelp to provide a little more contextual detail about the study site and region.

Line 165 - can the authors elaborate on what they mean by a phenomenological approach?

Line 180 - can the authors specify whether the WHO questionnaire was a translated version in Runyankore or in English?

Line 183 - what does "baccalaureate nurse" mean? Those with an undergraduate degree?

Line 186 - what were the purposeful sampling criteria?

Line 199 - please provide the interview guide as a supplementary file.

Line 214 - how was the point of saturation determined?

Line 256 - please describe the process of participant validation.

Line 270 - what do the authors mean by "this produced subjective results".

RESULTS

Line 276-7 - this first sentence is not required.

The level of detail provided in Table 1 seems likely to be identifiable. Rather than listing individual participant demographics, I suggest reporting the mean age and gravidity across the sample and percentages for marital status, trimester, occupation and education.

Overall, the analysis is at quite a surface level and the themes seem to represent the topic guide questions more than the content from participants. For example, the first theme (Coercive behaviour in intimate relationships) groups together experiences of violence, women's self-blame, and mental health impacts - which ought perhaps to be sub-themes or perhaps the top-level themes could be more interpretive. Similarly, "Drivers of IPV" includes content about women's hopelessness (e.g. line 379-380) and being trapped with no options for escape, which are rather different to drivers.

A possible structure for the analysis could be findings at the levels of the individual (man and woman), community, and wider society. At present the separation of these is not very easy to follow.

As I mention above, coercion is used interchangeably with violence and it should not be. E.g. Stepping on the gravid abdomen is not coercion, it is physical assault or violence. Being violated due to pregnancy is not reproductive coercion, it is rape/sexual assault/violence.

Line 319 - this quote has already been presented above.

Line 355 - "gave birth to physical violence" - is this an intentional play on words?

Line 417 - praying to break through negative thoughts or praying for IPV to end?

Line 440 Barriers to seeking support - the structure of this sub-section would benefit from re-organisation. The authors summarise a lot of points in the first, dense, paragraph and then list three quotations one after the other. It would help to walk the reader through the different sub-themes. In this first paragraph the authors also do not highlight the point about community reinforcement of abuse - e.g. the points at lines 454 and 464 - these show deep community acceptance/normalisation of IPV and active efforts by those who could support women to discourage their help-seeking.

:ine 445 - "overdependence" sounds like it is blaming the woman. Perhaps "dependence" would suffice?

Line 480 - "abandoned their first violent relationship" - again sounds like an an error by the woman - perhaps "fled a previous violent relationship"?

DISCUSSION

The Results, Discussion and Conclusions have some repetition that could be removed to make the paper more concise.

Line 515 - when referencing other literature, the authors do not acknowledge substantial literature on perinatal IPV from other parts of sub-Saharan Africa, especially Ethiopia and South Africa.

I repeat my comment that "coercion" should not be used interchangeably with violence or abuse.

Line 520 - While this sentence is uncontroversial, it does not respond directly to the findings of the study.

Line 533 - a natural recommendation from this finding would surely be that men require targeted education to address the root causes of hostility towards pregnant women and the unborn and help them find alternative outlets for their emotions than violence.

Line 556 - the women expressed that men prevented them from accessing vocational opportunities which were available - so it seems that the recommendation should be about working with men around the acceptability of women's empowerment in addition to any interventions to offer skills and income generating activities.

Line 566 I am not sure that the paper found that intergenerational violence was "one of the main" drivers.

Line 575 - would the authors also advocate action at the community level? There have been high profile studies in Uganda (e.g. SASA!) which showed this to be effective.

Line 576 - I would have thought that helplessness is less of a driver and more of a consequence that then reinforces the persistence of IPV. The description in this paragraph seems to somewhat repeat the text above about economic dependence.

Line 586 - again, while men's involvement in childcare is to be encouraged, I am not sure that this recommendation arises from the study's findings.

Line 604 - it should be discussed here that the people women shared their problems with were often not supportive and sometimes actively encouraged them to tolerate abuse - and link this to the many studies finding the same in sub-Saharan Africa and elsewhere.

Line 605 - sharing with church leaders is not mentioned in the Results.

Line 613 - the sample who took part in this study were, by definition, known in ANC to be experiencing IPV. So this sub-group of women would not represent the group of women who do not disclose abuse. So I am not sure that the findings of this study do contradict the Iranian study, necessarily - there may be many women out there who the ANC providers do not know about.

Line 623 "passive coping strategies" - I wonder if a better phrase is "accommodating" abuse? Women adapt their behaviour as a survival tactic to avoid provoking a volatile partner's anger.

Line 648 - there are other reasons midwives may not inquire, including their own patriarchal norms and personal experiences of IPV victimsation.

Line 653 - it would be helpful in the methods to clearly state what (if any) training on IPV is currently provided to ANC providers in the study context.

LIMITATIONS

Line 677 - qualitative research does not set out to be generalisable to other contexts so this is not a limitation of the design.

However, the authors have not commented on the data being 6 years old and the many contextual changes that will have taken place in that time. It would help to briefly explain in the methods why the analysis has been conducted several years later and to acknowledge contextual differences between now and 2019 in the Limitations section.

A strength may be that the study is conducted in a region of Uganda less well-represented in existing research - is that the case?

MINOR POINTS

Data is plural of datum so should be treated as plural throughout e.g. line 37 should say "data were..."

I suggest stating "years" after all age ranges e.g. line 181

Line 290, 292 - Themes are created interpretively by the analysers - so it is questionable whether the term "emerge" is applicable. The authors might want to use other terms e.g. "we identified x themes".

The term "accurate" is used several times in the Methods but this is a rather a positivist term.

A careful grammar edit is needed for the small number of errors e.g. Line 570 - "what they have ever seen their parents' doing".

**Do you want your identity to be public for this peer review?** For information about this choice, including consent withdrawal, please see our Privacy Policy

Reviewer #4: **Yes: ** Dr Roxanne C Keynejad

---

## [Decision Letter · Decision Letter 4]

23 Jul 2025

PGPH-D-23-02096R4

Intimate Partner Violence Experience, support seeking and Coping Strategies among Pregnant Women in Southwestern Uganda

Dear Dr. Katushabe,

Thank you for submitting your manuscript to PLOS Global Public Health. After careful consideration, we feel that it has merit but does not fully meet PLOS Global Public Health’s publication criteria as it currently stands. Therefore, we invite you to submit a revised version of the manuscript that addresses the points raised during the review process.

The manuscript has been evaluated by a previous reviewer, and their comments are available below.

The reviewer feels that many of their comments have been addressed but have additional concerns. The reviewer would like further explanation regarding the traditional roles mentioned in the abstract and clarification on participant access to “movie watching” as a coping strategy. They have also provided many suggestions to improve the grammar, sentence structure, and readability of the manuscript.

Could you please carefully revise the manuscript to address all comments raised?

We look forward to receiving your revised manuscript.

Kind regards,

Katherine Demi Kokkinias, Ph.D.

Staff Editor

Journal Requirements:

Additional Editor Comments (if provided):

Reviewers' comments:

Reviewer's Responses to Questions

**Comments to the Author**

Reviewer #4: (No Response)

publication criteria?

Reviewer #4: Yes

3. Has the statistical analysis been performed appropriately and rigorously?

Reviewer #4: N/A

4. Have the authors made all data underlying the findings in their manuscript fully available (please refer to the Data Availability Statement at the start of the manuscript PDF file)?

Reviewer #4: No

5. Is the manuscript presented in an intelligible fashion and written in standard English?

Reviewer #4: Yes

Reviewer #4: Thank you to the authors for their comprehensive response to my previous review. My comments have largely been addressed and I think the current version of the manuscript is much clearer. There remain some minor points which need to be addressed prior to publication. My line references refer to the version with changes tracked.

Research issues:

METHODS

Line 231 can the purposive sampling techniques be briefly described. Which participant characteristics were being sought?

DISCUSSION

Line 901 the authors have replaced the sentence suggesting that lack of generalisability is a limitation with an almost identical sentence. As I said, previously, this is not a valid critique of qualitative methods as this is not their purpose.

Minor points:

ABSTRACT

Line 46: I suggest "their households" or "the household".

Line 47: Differences in tranditional roles and norms between what and what? Between the man and woman? Between his upbringing and modern Ugandan women's lives?

INTRODUCTION

Line 71: "years" is still not used uniformly following age ranges.

Line 125: watching movies is, I assume, a coping strategy available only to women with a certain degree of affluence? Is it an appropriate example for this study sample?

Line 156 prevent and treat mental health conditions?

Line 162 I suggest deleting "However".

METHODS

Line 280 I would suggest "would be unlikely to provide..." rather than "would not provide".

Line 347 I suggest deleting "objective" as this term is not in keeping with the qualitative approach used.

RESULTS

Table 2: this is much more confidential, thank you. Please however add units where appropriate, e.g. "Mean age (years)" and "Mean gravidity (number of pregnancies".

Line 369: The authors have deleted "emerged" from the text but not the heading.

Line 533 I suggest deleting "were" in "some were refrained from earning..."

DISCUSSION

Line 828 Perhaps additional "intervention" research rather than "quantitative" research?

Line 904-5 "less well-represented subject" - perhaps "under-represented" is clearer?

GENERAL

A grammar check is needed. Issues include the following:

The tense - past tense is used often in the Introduction where present tense would be more natural.

The need for clarity following corrections, e.g. "making it makes it difficult" (line 79).

Lack of agreement:

- "woman's age, women's decision making ability" - I suggest agreement i.e. both being "woman" or "women" (line 86)

- "Uganda, Tanzania, Ethiopia and in New Jersey" - I suggest replacing New Jersey with the United States as all the other examples are countries (line 824).

- "data was gathered" (line 896) should be "data were..."

Punctuation (e.g. "a victims' isolation" (line 111).

Fragments in the text (e.g. "On the other hand," line 132, "Pregnant women..., line 313-4 - this sentence is agrammatical)

Use of articles e.g. "Although the Uganda's..." (line 159).

Inappropriate use or lack of capitals (e.g. "some IPV victims", line 134; "conducted In-depth interviews", line 257).

Spelling: line 360 "prime gravida" should, I think, be "primigravidae"?

Phrasing e.g. line 408 "their women" - this is language which implicitly characterises women as possessions. I suggest "their partner".

Consistency of acronyms e.g. "intimate partner violence" instead of IPV (line 471).

Sentence structure e.g. "the individuals with whom women confided their problems with" (line 819).

**Do you want your identity to be public for this peer review?** For information about this choice, including consent withdrawal, please see our Privacy Policy

Reviewer #4: **Yes: ** Dr Roxanne C Keynejad

---

## [Decision Letter · Decision Letter 5]

24 Sep 2025

PGPH-D-23-02096R5

Intimate Partner Violence Experience, support seeking and Coping Strategies among Pregnant Women in Southwestern Uganda

Dear Dr. Katushabe,

Thank you for submitting your manuscript to PLOS Global Public Health. After careful consideration, we feel that it has merit but does not fully meet PLOS Global Public Health’s publication criteria as it currently stands. Therefore, we invite you to submit a revised version of the manuscript that addresses the points raised during the review process.

We look forward to receiving your revised manuscript.

Kind regards,

Lina Taing

Academic Editor

Journal Requirements:

1. We would like to request copyediting.

Additional Editor Comments (if provided):

Reviewer #4: Thanks you for your comprehensive attention to their editorial comments and has accepted your paper for publication without further peer review. Only very minor revisions addressing grammar and syntax are needed from this reviewer.

Reviewer #6: Requests minor revisions addressing methodological rigor and structural clarity. Key issues include: clarifying the relationship between the 25-participant qualitative study and the larger 283-participant project; explaining participant selection methods and consent procedures for minors; strengthening the introduction by better articulating the research gap and novel contribution; restructuring themes to reflect genuinely emergent findings rather than predetermined categories; addressing overlapping themes and potential researcher bias; reframing stated "strengths" as actual limitations (particularly regarding generalizability from a single clinic); and adding missing sections on research implications. The reviewer emphasizes the need to distinguish between deductive analysis based on study aims and true inductive thematic emergence, while also improving transitions and clarifying key definitions throughout the manuscript.

Reviewers' comments:

Reviewer's Responses to Questions

**Comments to the Author**

Reviewer #4: All comments have been addressed

Reviewer #6: (No Response)

publication criteria?

Reviewer #4: Yes

Reviewer #6: Yes

3. Has the statistical analysis been performed appropriately and rigorously?

Reviewer #4: N/A

Reviewer #6: No

4. Have the authors made all data underlying the findings in their manuscript fully available (please refer to the Data Availability Statement at the start of the manuscript PDF file)?

Reviewer #4: No

Reviewer #6: Yes

5. Is the manuscript presented in an intelligible fashion and written in standard English?

Reviewer #4: Yes

Reviewer #6: Yes

Reviewer #4: Thank you to the authors for their comprehensive edits to this manuscript. I noted only some very minor points which do not require peer review once addressed. Line references refer to the tracked changes version.

Line 70 aged 15 years "and over"?

Line 151 it looks like a word or two is needed at the start of this sentence.

Line 171 I suggest changing 1 back to "one" month

Line 299 would "sub-themes" be preferable to "categories"?

Line 482 I believe "chose" (past tense) would be better than "choose".

Line 580 "keep and calm" should I think be "and keep calm".

Line 599 "further implementation research..." As this sentence is about exploring impacts on women I suggest deleting "implementation" as this would only have relevance for an intervention focus.

Reviewer #6: uploaded

**Do you want your identity to be public for this peer review?** For information about this choice, including consent withdrawal, please see our Privacy Policy

Reviewer #4: **Yes: ** Dr Roxanne Keynejad

Reviewer #6: No

---

## [Decision Letter · Decision Letter 6]

16 Dec 2025

Intimate Partner Violence Experience, support seeking and Coping Strategies among Pregnant Women in Southwestern Uganda

PGPH-D-23-02096R6

Dear Ms. Katushabe,

We are pleased to inform you that your manuscript 'Intimate Partner Violence Experience, support seeking and Coping Strategies among Pregnant Women in Southwestern Uganda' has been provisionally accepted for publication in PLOS Global Public Health.

Best regards,

Julia Robinson

Executive Editor

Reviewer Comments (if any, and for reference):

Reviewer's Responses to Questions

**Comments to the Author**

Reviewer #5: All comments have been addressed

publication criteria?

Reviewer #5: Yes

3. Has the statistical analysis been performed appropriately and rigorously?

Reviewer #5: N/A

4. Have the authors made all data underlying the findings in their manuscript fully available (please refer to the Data Availability Statement at the start of the manuscript PDF file)?

Reviewer #5: Yes

5. Is the manuscript presented in an intelligible fashion and written in standard English?

Reviewer #5: Yes

Reviewer #5: No further comments

**Do you want your identity to be public for this peer review?** For information about this choice, including consent withdrawal, please see our Privacy Policy

Reviewer #5: **Yes: ** Deda Ogum
